# Minimal effective dose of bedaquiline administered orally and activity of a long acting formulation of bedaquiline in the murine model of leprosy

**Aurélie Chauffour** [1], **Nacer Lounis**[2], **Koen Andries**[2], **Vincent Jarlier**[1,3], **Nicolas Veziris**[1,4], **Alexandra Aubry**[1,3] *

**1** Sorbonne Université, INSERM, Centre d'Immunologie et des Maladies Infectieuses (Cimi-Paris), Paris, France, **2** Janssen Pharmaceutica, Beerse, Belgium, **3** AP-HP. Sorbonne Université, Hôpital Pitié-Salpêtrière, Centre National de Référence des Mycobactéries et de la Résistance des Mycobactéries aux Antituberculeux, Paris, France, **4** APHP. Sorbonne-Université, Department of Bacteriology, Saint-Antoine Hospital, Paris, France

* alexandra.aubry@sorbonne-universite.fr

## Abstract

### Background

Bedaquiline (BDQ), by targeting the electron transport chain and having a long half-life, is a good candidate to simplify leprosy treatment. Our objectives were to (i) determine the minimal effective dose (MED) of BDQ administered orally, (ii) evaluate the benefit of combining two inhibitors of the respiratory chain, BDQ administered orally and clofazimine (CFZ)) and (iii) evaluate the benefit of an intramuscular injectable long-acting formulation of BDQ (intramuscular BDQ, BDQ-LA IM), in a murine model of leprosy.

### Methodology/Principal findings

To determine the MED of BDQ administered orally and the benefit of adding CFZ, 100 four-week-old female nude mice were inoculated in the footpads with $5\times10^3$ bacilli of *M. leprae* strain THAI53. Mice were randomly allocated into: 1 untreated group, 5 groups treated with BDQ administered orally (0.10 to 25 mg/kg), 3 groups treated with CFZ 20 mg/kg alone or combined with BDQ administered orally 0.10 or 0.33 mg/kg, and 1 group treated with rifampicin (RIF) 10 mg/kg. Mice were treated 5 days a week during 24 weeks.

To evaluate the benefit of the BDQ-LA IM, 340 four-week-old female swiss mice were inoculated in the footpads with $5\times10^3$ to $5\times10^1$ bacilli (or $5\times10^0$ for the untreated control group) of *M. leprae* strain THAI53. Mice were randomly allocated into the following 11 groups treated with a single dose (SD) or 3 doses (3D) 24h after the inoculation: 1 untreated group, 2 treated with RIF 10 mg/kg SD or 3D, 8 treated with BDQ administered orally or BDQ-LA IM 2 or 20 mg/kg, SD or 3D.

Twelve months later, mice were sacrificed and *M. leprae* bacilli enumerated in the footpad.

**Data Availability Statement:** Raw datas are available on the platform recherche.data.gouv.fr

under the DOI number https://doi.org/10.57745/RMFZJQ.

**Funding:** This work was supported by Janssen (Grants ICD#1071584 and ICD#975393 to the research unit for supplies (AA, AC, VJ, NV) and salary of NL and KA).

**Competing interests:** We have read the journal's policy and the authors of this manuscript have the following competing interests: NL is an employee of Janssen and KA has retired from Janssen; they participated to the methodology design and writing of the manuscript. AA, AC, VJ, NV have no conflict of interest to declare except funding aforementioned.

All the footpads became negative with BDQ at 3.3 mg/kg. The MED of BDQ administered orally against *M. leprae* in this model is therefore 3.3 mg/kg. The combination of CFZ and BDQ 10-fold lower than this MED did not significantly increase the bactericidal activity of CFZ. The BDQ-LA IM displayed similar or lower bactericidal activity than the BDQ administered orally.

## Conclusion

We demonstrated that the MED of BDQ administered orally against *M. leprae* was 3.3 mg/kg in mice and BDQ did not add significantly to the efficacy of CFZ at the doses tested. BDQ-LA IM was similar or less active than BDQ administered orally at equivalent dosing and frequency but should be tested at higher dosing in order to reach equivalent exposure in further experiments.

## Author summary

The current multidrug therapy is effective against leprosy but remains long and difficult to observe for patients supporting the need of monthly -based treatment. Bedaquiline (BDQ), a diarylquinoline with a long half-life, is a candidate drug to shorten leprosy treatment by targeting the electron transport chain and inhibiting the ATP synthesis. In this work, we demonstrated that (i) the minimal effective dose of BDQ administered orally against *M. leprae* is 3.3 mg/kg, (ii) BDQ did not add significantly to the efficacy of CFZ at the doses tested, and (iii) BDQ long acting formulation was similar or less active than BDQ administered orally at equivalent dosing and frequency but should be tested at higher dosing in order to reach equivalent exposure in further experiments.

## Introduction

Leprosy remains a major health problem worldwide despite being one of the oldest infectious diseases, reported for more than 2000 years. The leprosy elimination goal as a public health problem set by the World Health Organization, aiming for a global prevalence rate of < 1 patient in a population of 10,000, was achieved in 2000, but up to 200,000 new cases are still reported each year [1]. The worldwide use of leprosy drugs starting in the 1980s and their access at no cost for patients since 1995 were tremendous in the ability to achieve leprosy elimination [2]. Nowadays, the WHO global strategy targets zero leprosy by 2030, but this goal has been hindered by sharp reduction of leprosy case detection during 2020–2021 due to the Covid-19 pandemic [3]. As with other bacteria of medical interest, antimicrobial resistance is observed in the causative agent *Mycobacterium leprae* in several parts of the world, despite multidrug therapy being the recommended standard leprosy treatment to avoid resistance selection since 1982. The first treatment of leprosy, consisting of a monotherapy of dapsone, led to the emergence of drug-resistance [4]. Despite the addition of rifampicin (RIF) in the 1960s, drug-resistant strains quickly emerged [5]. Moreover, the length of the treatment leads to a poor compliance by patients and may favor the emergence of resistant strains. Therefore, to simplify and to facilitate the direct observation of treatment, a shorter, fully supervisable, monthly-administered multidrug regimen for leprosy is highly desirable [6]. Finally, in addition to patients whose *M. leprae* isolates are resistant to RIF, special regimens are also required for individual patients who cannot take RIF because of allergy, concomitant drug interaction or intercurrent disease such as chronic hepatitis.

In 2005, a newly discovered class of antibiotics, the diarylquinoline, was reported to be highly bactericidal against *M. tuberculosis* in mice and later in the mouse models for *M. leprae* [7,8]. The lead compound, bedaquiline (BDQ), also called R207910 or TMC207, inhibits an enzyme belonging to the electron transport chain, the ATP synthase, by binding to the subunit c of the enzyme, leading to a decrease in bacterial metabolism. The bactericidal activity of BDQ administered orally against *M. leprae* observed in mice is similar to that of moxifloxacin and RIF supporting the launch of a clinical trial aiming at evaluating BDQ efficacy in multi bacillary (MB) leprosy [9]. BDQ administered orally is currently the only new drug under clinical trial for leprosy treatment [10].

Interestingly, *M. leprae* does not possess all the proteins along the electron transport chain [10] suggesting that associating inhibitors acting at different enzymes belonging to it may act synergistically and may display strong bactericidal anti-leprosy activity. Clofazimine (CFZ), whose mechanism of action is not fully understood, targets the electron transport chain at the level of the menaquinone. CFZ has been shown to be effective against leprosy [11–19]. The anti-leprosy activity of the association of the two drugs acting on electron transport chain, e.g. BDQ and CFZ, deserve to be evaluated. In addition, the genes encoding the *M. tuberculosis* MmpS5-MmpL5 efflux pump and repressor (*mmpR*, *Rv0678*) whose mutations are implicated in BDQ resistance are absent in *M. leprae* where it might contributes to a higher potency of bedaquiline against *M. leprae* compared to *M. tuberculosis* [10,20].

Two key properties of drugs administered in LAI (Long Acting Injectable) formulations are low aqueous solubility to preclude the rapid dissolution and release of the active drug substance, and a reasonably long pharmacokinetic (PK) elimination half-life, *i.e.*, slow clearance from the body. For an antimicrobial, another desired property is high potency, negating the need for high concentrations in the blood and allowing lower drug doses to be injected. Bedaquiline seems to be highly potent against *M. leprae* [9] at a lower dose than against *M. tuberculosis*. It has high lipophilicity (logP, 7.3), and a long half-life (about 24 h, functionally or effectively) which makes it suitable for use in an LAI formulation [21]. The efficacy of LAI BDQ has been already demonstrated in a latent tuberculosis infection mouse model [21–23]. Due to the very slow doubling time of *M. leprae*, a unique administration of LAI BDQ could also be considered.

In our work, we aimed to (i) determine the minimum effective dose (MED) of the BDQ administered orally, (ii) evaluate the benefit of combining BDQ administered orally and CFZ, and (iii) evaluate the benefit of a BDQ long-acting formulation in a murine model of leprosy.

## Methods

### Ethics statement

The experimental project was favorably evaluated by the ethics committee n°005 Charles Darwin localized at the Pitié-Salpêtrière Hospital. Clearance was given by the French Ministry of Higher Education and Research under the number APAFIS#9575–2017030114543467 v3. Our animal facility received the authorization to carry out animal experiments (license number D75-13-08). The persons who carried out the animal experiments had followed a specific training recognized by the French Ministry of Higher Education and Research and follow the European and the French recommendations on the continuous training.

### Materials

In both experiments, mice were infected with a *M. leprae* THAI53 strain. This strain was fully susceptible to the common antileprosy drugs (*i.e.* RIF, dapsone, CFZ and fluoroquinolones) [24]. The suspension used to inoculate mice was prepared from mice already infected by this

isolate one year earlier. Shepard and Mac Rae's method was used to prepare the suspension [25]. Briefly, the tissue from the footpads was aseptically removed and then grinded by using a GentleMacs Octo Dissociator (Miltenyi) under a volume of 2 ml of Hanks' balanced salt solution. Ten μl of the prepared suspension were taken to perform a Ziehl-Neelsen staining to count *M. leprae* Acid Fast Bacilli (AFB). Suspensions needed to inoculate mice were then further diluted in Hanks' balanced salt solution.

Respectively four-week-old nude (NMRI-*Foxn1*$^{nu/nu}$) for the determination of the minimal effective dose of BDQ administered orally and swiss mice for the evaluation of the BDQ-LA IM were purchased from Janvier Labs, Le Genest Saint Isle, France. The nude mice model aims to mimic low-immunity leprosy and the swiss, high-immunity leprosy; therefore those two models will respectively represent multibacillary vs paucibacillary leprosy.

RIF and CFZ were purchased from Merck, France; BDQ administered orally and BDQ-LA IM were provided by Johnson and Johnson, Belgium.

## Infection of mice with *M. leprae* and treatment

**First experiment: Determining the minimal effective dose of BDQ administered orally against *M. leprae* and the contribution of CFZ when combined with BDQ.** We adapted the continuous method to determine the MED of BDQ administered orally [26]. The MED is defined as the lowest dose of a drug that inhibits the growth of *M. leprae*, *i.e.* corresponding to the group where all the mice footpads remain negative. One hundred 4-week-old female nude mice were infected in the left hind footpad with 0.03 ml of the *M. leprae* isolate THAI53 according to Shepard's method [27] with an inoculum of $5\times10^3$ AFB/ footpad. Mice were then randomly allocated into one untreated control group and 9 treated groups of 10 mice each: RIF 10 mg/kg, BDQ administered orally 0.10, 0.33, 1, 3.3 or 25 mg/kg, CFZ 20 mg/kg and combinations of BDQ administered orally 0.10 or 0.33 mg/kg and CFZ 20 mg/kg. Treatment was given one month after inoculation, five days a week during 24 weeks by oral gavage under a volume of 0.2 ml per mouse. The RIF and CFZ dosages used in our study are the current doses used in the murine model of leprosy [13,28].

**Second experiment: Comparing the bactericidal activity of BDQ administered orally and BDQ-LA IM against *M. leprae*.** We used the proportional bactericidal method that allows to measure the bactericidal activity of a compound [29]. Three hundred and forty 4-week-old female swiss mice were infected in the left hind footpad with 0.03 ml of the *M. leprae* isolate according to Shepard's method [27]. Mice were inoculated with three different inocula of $5\times10^3$, $5\times10^2$, $5\times10^1$ AFB/ footpad except for the untreated control which was also inoculated with one $5\times10^0$ additional group. Mice were randomly allocated into one untreated control group and 10 treated groups of 10 mice each: RIF 10 mg/kg, BDQ administered orally 2 or 20 mg/kg, BDQ-LA IM 2 or 20 mg/kg. Treatment was given by oral gavage under a volume of 0.2 ml per mouse, except for the BDQ-LA IM which was injected intramuscularly under a volume of 0.012 ml per thigh and both thighs were injected at the same time. Treatment for all drugs was given as a single (SD) or three doses (3D) for BDQ administered orally 2 or 20 mg/kg and BDQ-LA IM 2 or 20 mg/kg and began the day after inoculation for the SD, and 4 and 8 weeks later for 3D.

**Assessment of the effectiveness of the treatment.** To permit multiplication of *M. leprae* to a detectable level, mice were held 12 months in the animal facility. Mice were then euthanized and tissues from their footpad were removed aseptically and homogenized under a volume of 2 ml of Hank's balanced salt according to the Shepard's method [27]. *M. leprae* bacilli were considered to have multiplied (*i.e.* survived the treatment) if those footpads were found to contain $\geq10^5$ acid-fast bacilli, regardless of the size of the inoculum.

## Statistical analysis

**First experiment: Determine the MED of BDQ administered orally against *M*. *leprae*.** A Fisher exact test was performed. A p-value <0.05 was considered to be statistically significant by standard evaluation. For multiple comparisons between the groups, Bonferroni's correction was applied, *i*.*e*., the difference would be significant at the 0.05 level only if the P value adjusted to the number of groups: 0.05/n in which n was defined as the number of primary comparisons. Thus, the corrected P was 0.05/10 = 0.005.

**Second experiment: Compare the bactericidal activity of BDQ administered orally and BDQ-LA IM against *M*. *leprae*.** The proportion of viable *M*. *leprae* after treatment was determined from the infectious dose required to show multiplication in 50% of the inoculated mice. The significance of the differences between the groups was calculated by the Spearman and Kärber method [30]. A p-value <0.05 was considered statistically significant by standard evaluation. For multiple comparisons between the groups, Bonferroni's correction was applied, *i*. *e*., the difference would be significant at the 0.05 level only if the P value adjusted to the number of groups: 0.05/n in which n was defined as the number of primary comparisons. Thus, the corrected P was 0.05/11 = 0.0045.

## Results

### Minimal effective dose of BDQ administered orally (Table 1 and Fig 1)

After one year of observation, all footpads were positive in the untreated control group (mean of 8.13±0.26 $\log_{10}$ AFB per footpad), confirming the multiplication of *M*. *leprae*.

**Table 1. Multiplication of *M*. *leprae* organisms in nude mice to determine minimal effective dose of BDQ administered orally active against *M*. *leprae* and the benefit of the combination of CFZ and BDQ.**

| Treatment[a] | Positive footpad/ total footpad | Range of AFB/positive footpads (mean $\log_{10}$ AFB/ footpad standard deviation on positive footpads) | p value[b] |
|---|---|---|---|
| Untreated control | 10/10 | 7.79–8.67 (8.13±0.26) | / |
| RIF 10 mg/kg | 6/10 | 4.54–5.72 (NA[c]) | 0.08 |
| BDQ 0.10 mg/kg | 10/10 | 6.62–8.06 (7.39±0.50) | 1 |
| BDQ 0.33 mg/kg | 10/10 | 6.50–7.57 (7.02±0.40) | 1 |
| BDQ 1 mg/kg | 10/10 | 7.08–7.94 (7.56±0.32) | 1 |
| BDQ 3.3 mg/kg | 0/10 | NA[c] | 0.00001 |
| BDQ 25 mg/kg | 0/10 | NA[c] | 0.00001 |
| CFZ 20 mg/kg | 3/10 | 4.54 (NA[c]) | 0.003 |
| BDQ 0.10 mg/kg + CFZ 20 mg/kg | 2/10 | 4.54–5.15 (NA[c]) | 0.0007 |
| BDQ 0.33 mg/kg + CFZ 20 mg/kg | 1/10 | 5.54 (NA[c]) | 0.00009 |

[a] treatment was given by oral gavage 5 days a week for 24 weeks beginning one month after inoculation

[b] comparisons of the proportion of mice with positive footpads of each treated group versus untreated control (a p-value <0.005 was considered to be statistically significant when applying Bonferroni's correction).

[c] not applicable

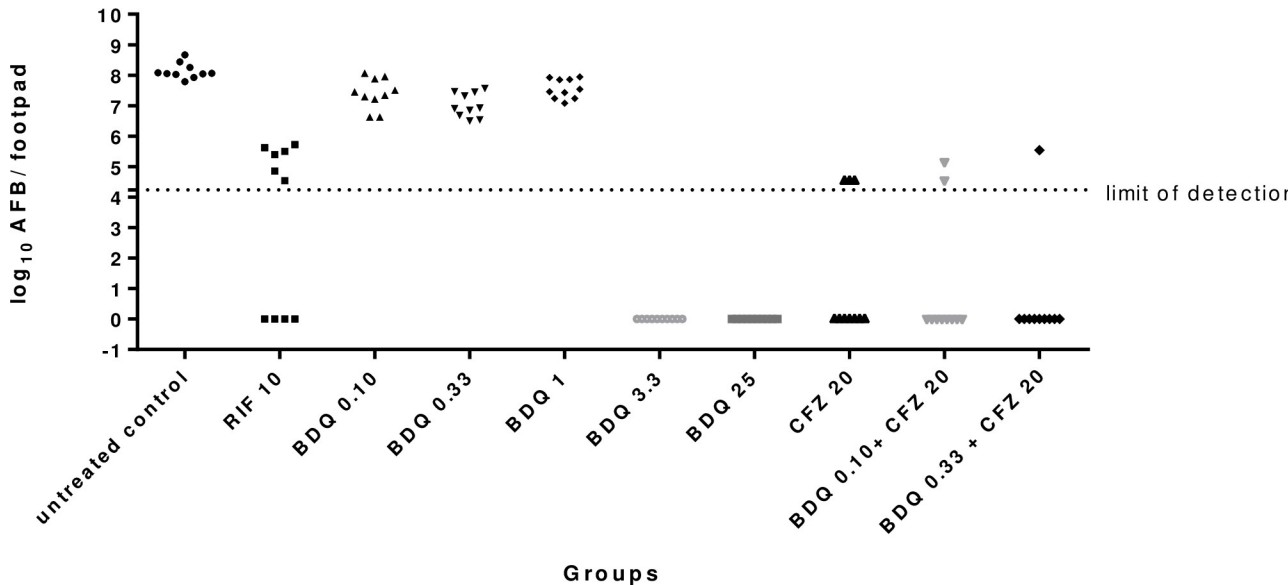

**Fig 1. Multiplication of *M. leprae* organisms in mice treated by BDQ administered orally and the benefit of the combination of CFZ and BDQ (each mouse footpad is taken as a data point and the dotted line indicates the threshold of detection of *M. leprae*).**

As compared to untreated control, RIF reduced the bacillary load since 4 mice had negative footpads after treatment (p = 0.08).

All mice footpads remained positive after treatment in the 3 lowest BDQ administered orally doses (0.10, 0.33 and 1 mg/kg) but they reduced the bacillary load by 1 log10 AFB as compared to the untreated control (p = 0.002, p = 0.00001, p = 0.0009 respectively). These 3 BDQ administered orally doses were also less bactericidal than RIF (10 mice remained positive after treatment in the 3 BDQ administered orally doses groups vs 6 mice positive in the RIF group) but it was not statistically significant (p = 0.08). On the other hand, all the footpads were negative in the groups treated with the 2 highest BDQ administered orally doses 3.3 mg/kg and the 25 mg/kg which was statistically significant regarding untreated control (p = 0.00001 for both groups). These groups displayed higher bactericidal activity than RIF 10 mg/kg but the difference was not statistically significant with Bonferroni's correction (p = 0.01 for both groups).

CFZ 20 mg/kg was bactericidal as compared to untreated control (p = 0.03) even if 3 mice remained positive after treatment. CFZ was as bactericidal as RIF with a p value not statistically different (p = 0.370). The combination of the 2 lowest BDQ administered orally doses (*i.e.* 0.10 or 0.33 mg/kg) with CFZ lead to a decrease in the number of AFB positive footpads as compared with CFZ but the reduction was not statistically significant (p = 1 and 0.58, respectively).

## Comparison of BDQ administered orally and BDQ-LA IM (Table 2)

From results in the untreated group, the proportion of viable *M. leprae* was estimated to be 2.75% of the total number of AFB inoculated. The percentage of viable bacilli killed under treatment ranged between 74.86% and 99.85% depending on the treatment group.

Compared to untreated control, the percentage of viable bacilli was smaller in all groups except BDQ 2 mg/kg SD either oral or LA IM; but the difference between the untreated group and the RIF 10 mg/kg SD was not significant after Bonferroni's adjustment. RIF 3D was significantly different from RIF SD (p<0.001).

**Table 2. Bactericidal activity against *M. leprae* THAI53 of bedaquiline administered orally and bedaquiline long-acting measured in Swiss mice by the proportional bactericidal method.**

| Treatment[a] | No. of footpads showing multiplication[b] of *M. leprae*/No. of footpads harvested, by inoculum | | | | % viable *M. leprae*[c] | % viable *M. leprae* killed by treatment[d] | p value[e] | p value[h] |
|---|---|---|---|---|---|---|---|---|
| | 5x10³ | 5x10² | 5x10¹ | 5x10⁰ | | | | |
| Untreated control | 10/10 | 8/10 | 8/10 | 2/10 | 2.753 | / | | |
| RIF 10 mg/kg SD[f] | 8/10 | 6/10 | 4/10 | - | 0.275 | 90.01 | 0.0046 | <0.001 |
| RIF 10 mg/kg 3D[g] | 0/10 | 0/10 | 0/10 | - | ≤0.004 | ≥99.85 | <0.001 | |
| BDQ 2 mg/kg SD | 9/10 | 7/10 | 6/10 | - | 0.692 | 74.86 | 0.075 | 1 |
| BDQ-LA IM 2 mg/kg SD | 9/10 | 8/10 | 5/10 | - | 0.692 | 74.86 | 0.069 | |
| BDQ 2 mg/kg 3D | 9/10 | 5/10 | 5/10 | - | 0.347 | 87.39 | 0.009 | 0.584 |
| BDQ-LA IM 2 mg/kg 3D | 8/10 | 6/10 | 3/10 | - | 0.219 | 92.04 | 0.0016 | |
| BDQ 20 mg/kg SD | 3/10 | 0/10 | 0/10 | - | 0.009 | 99.67 | <0.001 | 0.005 |
| BDQ-LA IM 20 mg/kg SD | 7/10 | 3/10 | 1/10 | - | 0.055 | 98.00 | <0.001 | |
| BDQ 20 mg/kg 3D | 0/10 | 0/10 | 0/10 | - | ≤0.004 | ≥99.85 | ≤0.001 | ≤0.014 |
| BDQ-LA IM 20 mg/kg 3D | 3/10 | 2/10 | 0/10 | - | 0.014 | 99.49 | <0.001 | |

[a] treatment was given by oral gavage, except for BDQ-LA IM which was administered intramuscularly

[b] *M. leprae* bacilli were considered to have multiplied if the harvest from a footpad yielded >10⁵ acid-fact bacilli

[c] the proportion of viable *M. leprae* surviving the treatment could be calculated by estimating the "most probable number" of viable organisms. However, the estimation of the MPN is based on the assumption that the organisms are distributed randomly in an inoculum; in the case of *M. leprae*, this assumption is probably untenable, therefore, the preferred alternative is to calculate the "median infectious dose (ID50)", i.e. the number of organisms required to infect 50% of the mice as allowed by the Spearman-Kärber method (it requires that the titration be carried out over a range of 100% to 0%). In immunocompetent mice, if the largest inoculum is 5X10³ *M. leprae* per footpad, a proportion of viable *M. leprae* as small as 0.00006 may be measured, then it is possible to calculate the proportion of viable *M. leprae* killed by the treatment by comparing the proportions of viable in tested and control mice. The significance of the differences between the groups was calculated by the Spearman and Kärber method [30]

[d] calculated from the comparison of the proportion of viable organisms between untreated controls and the treated group.

[e] each treated group was compared to the untreated group of mice

[f] drug was given under a single dose the day after infection

[g] drug was given three times (the day after infection, 4 and 8 weeks later)

[h] p value corresponds to the comparison of the 2 groups at the beginning of the line: RIF SD vs 3D, BDQ oral vs LA-IM at equivalent dosing and frequency

When comparing groups treated with 2 mg/kg BDQ administered orally or BDQ-LA IM at the same frequency (SD or 3D), a similar number of footpads remained positive in both groups (p>0.05). When comparing groups treated with 20 mg/kg BDQ administered orally or BDQ-LA IM at the same frequency (SD or 3D), more footpads remained positive in the groups treated with BDQ-LA IM and the difference were statistically significant but not after Bonferroni adjustments whatever the number of administrations between BDQ and BDQ-LA IM groups (p>0.0045) (Table 2).

## Discussion

The current length of the leprosy treatment remains a challenge [31]. During the last decades, few new antituberculous drugs were synthesized and even rarer are those active against *M. leprae*. In 2006, Ji et al. demonstrated the antileprosy activity of BDQ administered orally, a new diarylquinoline which was also active against other mycobacteria such as *M. tuberculosis* [6,7]. They showed that a single dose of 25 mg/kg BDQ was as effective as rifapentine, moxifloxacin, as well as RIF, which is currently the most powerful antileprosy drug. Nevertheless, the minimal effective dose of the BDQ administered orally is unknown despite knowing that it may enable developing a safe dose. In our present work, we determined that a dose as low as

3.3 mg/kg of BDQ administered orally displayed a bactericidal activity indistinguishable from a 25 mg/kg dose (Table 1 and Fig 1). The MED of 3.3 mg/kg is higher than the ≤1 mg/kg that was found to be the lowest active dose in an immunocompetent mouse model of leprosy [8] which can be explained by the lack of T-cell immunity, known to be important in mycobacterial diseases, in the nude mouse model used to determine the MED of BDQ administered orally in the present study. The MED of BDQ administered orally was not achieved in the immunocompetent mouse model because all the doses tested including the lowest one (1 mg/kg) were able to reach the limit of detection of *M. leprae* present in mouse footpads as enumerated by the AFB microscopy [8] suggesting a MED of ≤ 1mg/kg. The MED of 3.3 mg/kg obtained in the immunocompromised mouse model of leprosy is lower than that obtained with *M. tuberculosis* in an immunocompetent mouse model (6.5 mg/kg) [8] but the MED of BDQ administered orally against *M. tuberculosis* should be compared to the MED of BDQ administered orally against *M. leprae* determined in an immunocompetent mouse model (MED ≤ 1 mg/kg) suggesting at least a 6-fold lower MED of BDQ administered orally against *M. leprae* compared to that of *M. tuberculosis*.

It's important to mention that a single dose of 2 mg/kg BDQ administered orally (total dose = 2 mg/kg) in experiment 2 was able to kill 75% of the viable *M. leprae* bacilli while 6 months of 5 days per week treatment of 1 mg/kg (total dose = 120 mg/kg) in experiment 1 was not able to kill any bacilli present at start of treatment but was only able to slow down the replication of the bacilli when compared to untreated mice. The main differences between experiments 1 & 2 are the use of immunocompromised mice and initiation of treatment one-month post-infection in experiment 1 and the use of immunocompetent mice and initiation of treatment the day after infection in experiment 2. The use of immunocompromised mice without T cell immunity allows a much better replication of *M. leprae* and a much higher rate of viable bacilli while the replication rate and the rate of viable bacilli are much lower in immunocompetent mice which partially explain the differences in the effectiveness of the 2 regimens.

The treatment of leprosy needs to be based on a drug-combination to avoid the emergence of resistant-strains. BDQ targets the electron transport chain, which is also the target of the classical anti-leprosy drug CFZ [12,13]. Combining low doses of BDQ that were ineffective alone with CFZ enabled to reduce the numbers of AFB positive footpads (Table 1), however, BDQ did not add significantly to the efficacy of CFZ at the tested doses. These 2 combinations were slightly more bactericidal than RIF. Despite being not statistically significant, this result may suggest that a combination of drugs targeting the electron transport chain may be highly bactericidal against leprosy. Further experiments should be designed and performed in nude mice rather than in Swiss, since the much larger numbers of *M. leprae* viable organisms in nude mice than in immunocompetent Swiss mice may permit more accurate differentiation among various levels of bactericidal activities.

One of the main characteristics of *M. leprae* is its long doubling time (*i.e.* 14 days) suggesting an active drug with a long half-life would be a good choice against this bacterium. The drug must be effective at a low dose and under an intermittent administration, conditions that are currently sought for the treatment of leprosy. A new formulation of BDQ, called BDQ-LA IM, was found to be active in a mouse model of latent tuberculosis [22]. Despite the potential challenges of introducing an injectable formulation in the field, its improved pharmacokinetics properties compared to BDQ administered orally may allow reduction in the duration of the treatment and therefore increase adherence of patients to the treatment. The BDQ-LA IM formulation was tested at two dosages, 2 or 20 mg/kg, and with two frequencies (one, SD or three doses, 3D). Our results showed that at equivalent dosing and frequency, BDQ administered was similar, or even more active than LA-IM in our murine model of leprosy (Table 2). A possible explanation why the 20 mg/kg was less effective when administered as a LA injection

rather than orally is that the doses tested in our experiment were too low for LA-IM BDQ. Indeed, in the studies evaluating the BD-LA IM in tuberculosis higher doses were used and it was shown that BDQ LA-IM 160 mg/kg generated a $C_{max}$ equivalent to that of 30 mg/kg BDQ administered orally. In support of this hypothesis is the fact that in the present study when comparing 20 mg/kg LA-IM BDQ to 2 mg/kg BDQ administered orally at equivalent dosing, the injectable form was more active than the oral form. Consequently BDQ LA-IM should be further tested at higher dosing against leprosy.

In conclusion, we found that the MED of BDQ administered orally against *M. leprae* was 3.3 mg/kg and that the combination of CFZ and BDQ may improve the bactericidal activity. BDQ-LA IM was similar or less active than BDQ administered orally at equivalent dosing and frequency but should be tested at higher dosing in order to reach equivalent exposure in further experiments. These findings open the path to the design of shorter BDQ-based treatment deserving to be evaluated in leprosy.

## Author Contributions

**Funding acquisition:** Vincent Jarlier.

**Methodology:** Nacer Lounis, Koen Andries, Vincent Jarlier, Nicolas Veziris, Alexandra Aubry.

**Supervision:** Alexandra Aubry.

**Writing – original draft:** Aurélie Chauffour, Alexandra Aubry.

**Writing – review & editing:** Aurélie Chauffour, Nacer Lounis, Koen Andries, Vincent Jarlier, Nicolas Veziris, Alexandra Aubry.

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
