## [Decision Letter · Decision Letter 0]

23 Jun 2023

Dear Dr. Aubry,

Thank you very much for submitting your manuscript "Minimal effective dose of oral bedaquiline and activity of a long acting formulation of bedaquiline in the murine model of leprosy Bedaquiline for the treatment of leprosy" for consideration at PLOS Neglected Tropical Diseases. As with all papers reviewed by the journal, your manuscript was reviewed by members of the editorial board and by several independent reviewers. In light of the reviews (below this email), we would like to invite the resubmission of a significantly-revised version that takes into account the reviewers' comments. 

Dear Dr. Aubry and colleagues,

Thank you for your valuable submission to PLoS Neglected Tropical Diseases. The reviewers were enthusiastic about the work but felt there were numerous areas that required clarification and further analysis. Please pay particular attention to all comments (particularly those of reviewers #2 and #3) and upon resubmission, indicate your response to each comment and how it is addressed in the revised version.

In addition to the suggestions made by the reviewers, please make the following changes: 

line 33 and elsewhere: for 24 weeks

line 134 and elsewhere: in a volume of 0.2m Hanks' balanced salt solution

line 197: insert log10 after 8.13±0.26

line 227: controls

line 274: at the start of treatment

line 283: clarify that the targets of CFZ and BDQ are distinct; perhaps the target (at a different locus)

line 285: led to a reduction of the number of (instead of allowed to reduce)

line 294: delete an

For your information, you suggested Dr. James Krahenbuhl as a potential reviewer. Dr. Krahenbuhl would have been an excellent choice, but, unfortunately, he died in 2017.

We cannot make any decision about publication until we have seen the revised manuscript and your response to the reviewers' comments. Your revised manuscript is also likely to be sent to reviewers for further evaluation.

Sincerely,

Paul J. Converse

Academic Editor

Mathieu Picardeau

Section Editor

Dear Dr. Aubry and colleagues,

Thank you for your valuable submission to PLoS Neglected Tropical Diseases. The reviewers were enthusiastic about the work but felt there were numerous areas that required clarification and further analysis. Please pay particular attention to all comments (particularly those of reviewers #2 and #3) and upon resubmission, indicate your response to each comment and how it is addressed in the revised version.

In addition to the suggestions made by the reviewers, please make the following changes: 

line 33 and elsewhere: for 24 weeks

line 134 and elsewhere: in a volume of 0.2m Hanks' balanced salt solution

line 197: insert log10 after 8.13±0.26

line 227: controls

line 274: at the start of treatment

line 283: clarify that the targets of CFZ and BDQ are distinct; perhaps the target (at a different locus)

line 285: led to a reduction of the number of (instead of allowed to reduce)

line 294: delete an

For your information, you suggested Dr. James Krahenbuhl as a potential reviewer. Dr. Krahenbuhl would have been an excellent choice, but, unfortunately, he died in 2017.

Reviewer's Responses to Questions

**Key Review Criteria Required for Acceptance?**

**Methods**

-Are the objectives of the study clearly articulated with a clear testable hypothesis stated?

-Is the study design appropriate to address the stated objectives?

-Is the population clearly described and appropriate for the hypothesis being tested?

-Is the sample size sufficient to ensure adequate power to address the hypothesis being tested?

-Were correct statistical analysis used to support conclusions?

-Are there concerns about ethical or regulatory requirements being met?

Reviewer #1: The Objectives and Methods are correctly applied and clearly described. The numbering notation for the bacillary load injected into the mouse-foot-pads is not well displayed in the PDF version that was downloaded, although seems to be correct.

Reviewer #2: The rationale for using immunocompromised nude mice for the first study and immunocompetent mice for the second study could be better explained. Should they be considered as representing different phenotypes on spectrum of leprosy disease? How does the finding of a higher MED in immunocompromised mice compared to prior results in immunocompetent mice impact future development plans? For example, if the ongoing clinical trial evaluating a BDQ dose that is already lower than the current TB dose is successful? 

Could the authors comment on how well the CFZ dose of 20 mg/kg reflects exposures attained with the 50 mg daily dose used in leprosy treatment? 

For the statistical analysis of the first experiment, please clarify if the results from mice with AFB below the limit of detection were included and, if so, what value was assigned to them. Zero cannot be log-transformed and a value closer to the lower limit of detection would be more conservative and, arguably, more appropriate for statistical comparisons. 

The methods should describe how the MED is defined.

Reviewer #3: The methods meet the criteria for acceptance. It is very well written, clear and provides appropriate explanation citing key references across mouse foot pad and relevant drug sensitivity techniques.

**Results**

-Does the analysis presented match the analysis plan?

-Are the results clearly and completely presented?

-Are the figures (Tables, Images) of sufficient quality for clarity?

Reviewer #1: Clear and accurate.

Reviewer #2: If there is any information on the PK of these BDQ-LA doses used in the experiment, it should be included to better understand why the 20 mg/kg was less effective when administered as a LA injection rather than orally. 

Since the lower limit of detection is around 4.24 log10, it may not be accurate to compare between groups using log reduction values that assume that all mice without detectable AFB had zero AFB when the AFB could really lie anywhere in the range from zero to 4 log10. Suggest to avoid referring to log reductions that involve groups with AFB below the limit of detection. 

For readers unfamiliar with the proportional bactericidal model, it may be difficult to understand how the value for “% viable M. leprae” in Table 2 was derived. Please describe in more detail. For example, the authors could illustrate how the value of 2.753% was obtained for the untreated control in the results section.

Reviewer #3: The results section is well done.

**Conclusions**

-Are the conclusions supported by the data presented?

-Are the limitations of analysis clearly described?

-Do the authors discuss how these data can be helpful to advance our understanding of the topic under study?

-Is public health relevance addressed?

Reviewer #1: Clear and justified.

Reviewer #2: The interpretation of the results of treatment with BDQ and CFZ in combination requires some revision. Since the efficacy of CFZ monotherapy at the dose tested was far greater than that of BDQ monotherapy at the doses tested in the combination, the proper conclusion is not that CFZ adds to the efficacy of BDQ but rather than BDQ does not add significantly to CFZ at the doses tested. Further work to identify a BDQ dose that increases the efficacy of the combination when added to CFZ appears warranted.

Reviewer #3: The conclusion is drafted with good discussion and references to highlight findings, usage implications and future research. 

The first sentence of the discussion should be rephrased as duration of treatment is not the only challenge. Consider: "The current length of leprosy treatment remains a challenge." It may also be beneficial to cite Indian reports on leprosy treatment default or, more simply, the International Textbook of Leprosy's chapter on "Treatment of Leprosy" (https://internationaltextbookofleprosy.org/chapter/treatment). See the paragraph with default rates in some Indian studies ranging up to 34% in MB cases.

**Editorial and Data Presentation Modifications?**

Reviewer #1: Line 99: 'deserve' should be 'deserves'

Line 133: 'grinded' should be 'ground'

Reviewer #2: Line 99: “Rv0678” not “Rv6708”

Line 100: consider adding a more detailed description of the gene product beyond “inhibitor of efflux pumps” and provide the pseudogene name that is being referred to here, if possible

Line 101: does the same statement apply for clofazimine?

Line 112: delete “standard of”?

Line 114: suggest to replace “the oral BDQ” with “BDQ administered orally” or “orally administered BDQ”

Line 133: “was” not “were”

Lines 134 and 137: Hanks’ balanced salt solution

Line 141: please confirm whether the BDQ-LA IM is the same formulation that was studied in references 21-23 and confirm the concentration of BDQ in the formulation. If it is not the same, please describe the differences. 

Line 146: “combined with” not “combined to”

Lines 148-9 and elsewhere: suggest to use either “the Shepard method” or “Shepard’s method” but not “the Shepard’s method”

Line 149: and elsewhere, if 5.103 is the same as 5x103, the latter may be a more conventional way to represent it

Line 174: delete “were” OR change “in” to “if”?

Line 186-7: for readers unfamiliar with the model, it may be unclear what is meant by “50% infectious dose”. Perhaps the sentence would be clearer if rephrased with added detail as “The proportion of viable M. leprae after treatment was determined from the infectious dose required to show multiplication in 50% of the inoculated mice” or something similar.

Line 197: “…0.26 log10 AFB per footpad”

Lines 208-10: since the activity of CFZ was clearly greater than that of low-dose BDQ, it would be more appropriate here to ask if addition of BDQ increased the activity of CFZ, not the other way around. The answer appears to be “probably not”.

Line 214: suggest to replace “association” with “addition” or rephrase as “the combination of CFZ with BDQ”

Line 215: suggest to rephrase “during 24 weeks one month…” as “for 24 weeks beginning one month…”

Line 224: perhaps it would be clearer to rephrase the first sentence as “From results in the untreated group, the proportion of viable M. leprae was estimated to be 2.75% of the total number of AFB inoculated”? 

Line 251: when saying “few new molecules were synthesized”, please describe the therapeutic area you are referring to. For example, anti-infectives or anti-tuberculosis drugs? 

Line 258: consider replacing “similar to” with “indistinguishable from”. It is a subtle difference but would better reflect the fact that differences that all footpad CFU counts were below the limit of detection and any dose-related differences that may have been apparent with shorter durations of treatment were not assessable 

Line 296: would specify that BDQ-LA was active in a mouse model of latent TB

Line 309: given the preceding discussion, should it be specified that this MED is for immunocompromised mice?

Table 1: for the p value column, it would be helpful to indicate that this p value pertains to comparisons made on the basis of AFB/fp and not the proportion of footpads that are positive

Table 2: in the 5.102 column, the BDQ-LA IM 20 mg/kg SD group should have “3/10” not “3/0” 

Table 2: there appears to be an error in the RIF SD group with either the % viable M. leprae value of 0.432 or the % killed value of 90.01. Please check the values and confirm the accuracy of the statistical result.

Reviewer #3: Some sentences in the introduction should be rephrased or simplified for clarification. For example, "The next goal which remains to be achieved is to develop a strategy focusing on zero leprosy by the end of 2030." The WHO strategy for 2021-2030 is cited (3) and was developed during 2019-2020. The authors perhaps meant "The WHO global strategy targets zero leprosy by 2030. This goal has been hindered by (i) sharp reduction of leprosy case detection during 2020-2021 due to the pandemic (see other suggested citations below), (ii) ... this next part of the sentence becomes convoluted and could be a separate sentence on its own. 

WHO Weekly Epidemiological Record - every September issue includes an annual report on the previous year of global leprosy diagnoses. Therefore, 2021 data is reported in the 2022 edition - which adds more info relevant to your point. 

2021: https://reliefweb.int/report/world/weekly-epidemiological-record-wer-9-september-2022-vol-97-no-36-2022-pp-429-452-enfr

Line 85: concomitant drug interactions can be another reason 

Line 111: does BDQ have the longest half-life of current MDT and recognized secondline treatment drugs for leprosy? If so, that could be beneficial to state. 

The sentence on lines 283-284 should be rephrased for clarity. Consider: When combined with low doses of BDQ that were ineffective alone, CFZ enabled reduced numbers of AFB positive footpads.

**Summary and General Comments**

Reviewer #1: This is a straightforward paper with a message of interest to anyone involved in treating leprosy.

Reviewer #2: In this manuscript the authors describe two experiments to evaluate the activity of bedaquiline (BDQ) in mouse footpad infection models of leprosy. The first experiment evaluated a wide range of daily oral bedaquiline doses administered for 24 weeks beginning 1 month after infection in immunocompromised nude mice to determine the minimal effective dose (MED) of BDQ and included additional arms with CFZ alone and in combination with low dose BDQ. The second experiment evaluated two dose levels of BDQ (2 and 20 mg/kg) administered either orally or as a long-acting (LA) formulation, and either as a single dose (SD) or as 3 doses (3D) spaced 4 weeks apart, beginning immediately after infection in immunocompetent mice. In the first experiment, BDQ doses of 0.1, 0.33 and 1 mg/kg/d had modest but statistically significant effects on the AFB burden compared to no treatment, while doses of 3.3 and 25 mg/kg reduced the burden to levels below the limit of detection. Hence, the MED was determined to be 3.3 mg/kg. CFZ alone was also highly active and it was not apparent that adding BDQ at doses up to 0.33 mg/kg/d increased its activity. In the second experiment, BDQ showed dose-dependent and duration-dependent activity. At 2 mg/kg, results in SD arms were not statistically significantly different from no treatment, while 3D arms killed ~1 log10. Oral and LA forms had similar activity at this dose level. However, while both oral and LAI forms had significant killing effects at the 20 mg/kg dose level, the oral formulation was more active than the LA formulation at this dose level. The authors conclude that BDQ may be effective against leprosy at doses lower than those used for TB, that addition of CFZ may improve the efficacy of BDQ, and that LA BDQ formulations should be tested at higher doses in future experiments. 

The experiments yield novel and timely information since initial efforts are underway to evaluate the activity of daily oral BDQ in leprosy patients. However, the potential impact of this study on future preclinical and clinical development efforts is more difficult to judge. Some of the comments and questions provided are intended to help the reader to better understand the authors’ perspectives.

Reviewer #3: The novel information reported in this manuscript is critically important to improving best care options for leprosy cases. The work was done well in both competent and immunocompetent mouse models to best depict drug combination and regimen impact on M.leprae with clear connection to feasibility in patient care. The immunological and drug dynamics were well explained and referenced. There are only a few minor edits that could potentially improve clarity. Overall, it is a superb paper.

PLOS authors have the option to publish the peer review history of their article (what does this mean?). If published, this will include your full peer review and any attached files.

Reviewer #1: Yes: Paul Saunderson

Reviewer #2: No

Reviewer #3: No
---

## [Decision Letter · Decision Letter 1]

2 Oct 2023

Dear Dr. Aubry,

Thank you very much for submitting your manuscript "Minimal effective dose of bedaquiline administered orally and activity of a long acting formulation of bedaquiline in the murine model of leprosy" for consideration at PLOS Neglected Tropical Diseases. As with all papers reviewed by the journal, your manuscript was reviewed by members of the editorial board and by several independent reviewers. The reviewers appreciated the attention to an important topic. Based on the reviews, we are likely to accept this manuscript for publication, providing that you modify the manuscript according to the review recommendations. 

Dear Dr. Aubry,

Your revised manuscript has been reviewed. There remain a few editorial suggestions to be addressed before the manuscript can be accepted for publication.

Thank you for your patience and for submitting this work to PLoS Neglected Tropical Diseases.

Sincerely,

Paul J. Converse

Academic Editor

Mathieu Picardeau

Section Editor

Dear Dr. Aubry,

Your revised manuscript has been reviewed. There remain a few editorial suggestions to be addressed before the manuscript can be accepted for publication.

Thank you for your patience and for submitting this work to PLoS Neglected Tropical Diseases.

Reviewer's Responses to Questions

**Key Review Criteria Required for Acceptance?**

**Methods**

-Are the objectives of the study clearly articulated with a clear testable hypothesis stated?

-Is the study design appropriate to address the stated objectives?

-Is the population clearly described and appropriate for the hypothesis being tested?

-Is the sample size sufficient to ensure adequate power to address the hypothesis being tested?

-Were correct statistical analysis used to support conclusions?

-Are there concerns about ethical or regulatory requirements being met?

Reviewer #2: (No Response)

**Results**

-Does the analysis presented match the analysis plan?

-Are the results clearly and completely presented?

-Are the figures (Tables, Images) of sufficient quality for clarity?

Reviewer #2: (No Response)

**Conclusions**

-Are the conclusions supported by the data presented?

-Are the limitations of analysis clearly described?

-Do the authors discuss how these data can be helpful to advance our understanding of the topic under study?

-Is public health relevance addressed?

Reviewer #2: (No Response)

**Editorial and Data Presentation Modifications?**

Reviewer #2: Line 23: suggest to replace “adding” with “combining”

Lines 100-3: wouldn’t an inactivated mmpR gene be more likely to contribute to a lower potency of bedaquiline against M. leprae due to loss of the efflux repressor?

Line 141: consider to replace “compare” with “represent”

Line 216: suggest to replace “different” with “significant“

Line 228: suggest to replace “number” with “proportion”

Line 257: replace “probable” with “probably”

Line 258: “…titration be carried out…”

Line 259: replace “foot-pad” with “footpad”

Table 1: it is not clear what is meant by “median value on positive footpads” at the top of the 3rd column.

**Summary and General Comments**

Reviewer #2: The authors' responses and revisions have adequately addressed the issues raised in the prior reviews. A few minor editorial comments are provided above.

PLOS authors have the option to publish the peer review history of their article (what does this mean?). If published, this will include your full peer review and any attached files.

Reviewer #2: No

Figure Files:

Data Requirements:

Reproducibility:

References

---

## [Editor Report · Decision Letter 2]

28 Oct 2023

Dear Dr. Aubry,

We are pleased to inform you that your manuscript 'Minimal effective dose of bedaquiline administered orally and activity of a long acting formulation of bedaquiline in the murine model of leprosy' has been provisionally accepted for publication in PLOS Neglected Tropical Diseases.

Best regards,

Paul J. Converse

Academic Editor

Mathieu Picardeau

Section Editor

---

## [Editor Report · Acceptance letter]

20 Nov 2023

Dear Dr. Aubry,

We are delighted to inform you that your manuscript, "Minimal effective dose of bedaquiline administered orally and activity of a long acting formulation of bedaquiline in the murine model of leprosy," has been formally accepted for publication in PLOS Neglected Tropical Diseases.

Best regards,

Shaden Kamhawi

co-Editor-in-Chief

Paul Brindley

co-Editor-in-Chief
